# FHH Quick App Review: How Can a Quality Review Process Assist Primary Care Providers in Choosing a Family Health History App for Patient Care?

**DOI:** 10.3390/genes13081407

**Published:** 2022-08-08

**Authors:** Grant M. Wood, Sander van Boom, Kasper Recourt, Elisa J. F. Houwink

**Affiliations:** 1InfiniteHealth.care, Salt Lake City, UT 84103, USA; 24MedBox, 2321 JW Leiden, The Netherlands; 3Department of Public Health and Primary Care (PHEG), Leiden University Medical Centre, 2333 ZA Leiden, The Netherlands; 4National eHealth Living Lab (NELL), 2333 ZD Leiden, The Netherlands

**Keywords:** family health history (FHH), ISO/TS 82304-2, app assessment

## Abstract

Family health history (FHH) is a data type serving risk assessment, diagnosis, research, and preventive health. Despite technological leaps in genomic variant detection, FHH remains the most accessible, least expensive, and most practical assessment tool for assessing risks attributable to genetic inheritance. The purpose of this manuscript is to outline a process to assist primary care professionals in choosing FHH digital tools for patient care based on the new ISO/TS 82304-2 Technical Specification (TS), which is a recently developed method to determine eHealth app quality. With a focus on eHealth in primary care, we applied the quality label concept to FHH, and how a primary care physician can quickly review the quality and reliability of an FHH app. Based on our review of the ISO TS’s 81 questions, we compiled a list of 25 questions that are recommended to be more succinct as an initial review. We call this process the FHH Quick App Review. Our ‘informative-only’ 25 questions do not produce a quality score, but a guide to complete an initial review of FHH apps. Most of the questions are straight from the ISO TS, some are modified or de novo. We believe the 25 questions are not only relevant to FHH app reviews but could also serve to aid app development and clinical implementation.

## 1. Introduction

Primary care continues to implement, use, and trust more digital healthcare, affected by the increasing need for blended and coordinated care. eHealth-based technologies need to be optimally integrated into the daily clinical workflow in order to be sustainably utilized. Traditionally, many types of risk data are collected during healthcare diagnosis, management, and research. Family health history (FHH) is an exemplary data type for risk assessment, diagnosis, research, and preventative health, and is universally accessible [1]. However, the process of capturing and coordinating FHH data among healthcare providers is not yet fully digital.

### 1.1. FHH as First Assessment of Genetic Risk

Before clinical genetic tests became available, FHH was the only tool for assessing risks attributable to genetic inheritance, particularly for common diseases with genetic components. Despite technological leaps in genomic variant detection, FHH remains the most accessible, least expensive, and most practical assessment tool for this purpose, and remains the sole tool for those without consistent access to modern sequencing labs. FHH has been validated and recognized as the standard of care, with FHH statements (e.g., patient’s father had a heart attack at age 50) an optional component of electronic medical records (EMR) [2]. Going beyond this, current research is looking into the potential of combining polygenic risk scores and FHH [3]. This could be promising, showing the importance of including FHH in risk score prediction to enhance the accuracy of polygenic risk scores, particularly in diverse populations.

Certain clinical documentation requirements endorse the value of FHH because it changes management and motivates behavior change. Rather than being replaced by genetic testing, FHH enhances the quality of variant interpretation by modifying prior probability, particularly for variants of uncertain significance [4]. FHH could therefore serve as a crucial bridge between the observed phenomenon of inheritance and genomic variation.

### 1.2. Utility of FHH Not Fully Realized

Genetic testing trends indicate that FHH is more important than ever. FHH value in familial disorders is increasing, especially in cancer, advanced by better risk guidance [4]. The expanding genetic and genomic testing volume in general health care parallels new management options. On a patient level, earlier screening, family testing, and genetic counseling are some of the benefits. On a systems level, there is a renewed emphasis on managing health at the population level and on applying public health strategies to genetic and common diseases [5].

However, universal FHH use remains elusive [2,6]. Though quality FHH research and outreach exists, in many clinical contexts [2,7,8,9], patient, system, and provider factors persist. For example, modern families may be separated geographically and receive healthcare in different systems thus impeding FHH sharing both between health records and among family members. Accessibility of genetic testing is uneven in many locations, placing additional weight on FHH accuracy and rigorous application [10].

The effectiveness of FHH depends on user familiarity, frequency of use, the quality of its content, and the skill of its operator [11]. Unfortunately, these dependencies have not been systematically addressed, resulting in the full digital promise remaining unrealized [2]. Disciplined collection of FHH by clinicians remains low and recording is fragmented and differently formatted. Barriers to overcoming this are diverse and well documented [11]. Among them, time to collect and record, inadequate EMR user interfaces, lack of EMR interoperability, and inadequate provider and patient education are often cited [12,13,14]. Perhaps the use of FHH is still underutilized because of the complexity of the problem of effectively integrating and implementing FHH in daily medical care practice. Using reductionism in this manuscript might be a step forward to reducing the complex phenomena into more basic parts and help the primary care physician in assessing the quality and reliability of eHealth apps and utilizing FHH to its full potential [15].

### 1.3. More Examples

The knowledge of genetic variations associated with the risk of diverse diseases and response to medication will revolutionize clinical medicine. Pharmacogenetic innovations and knowledge will affect diverse populations, and thus lead to unbiased treatment through personalized medicine. Medication efficacy will be increased, and side effects can be decreased if pharmacogenomics information is utilized to its full potential. As a further example of unrealized clinical utility, family medication history (i.e., inefficacy and adverse reactions across family members throughout generations) should become a standard component of every patient’s family health history [16].

The lack of robust FHH data and limited sharing also hamper the ability of public and preventive health programs to target screening for inherited diseases for those at the highest risk [5]. The net result is an ineffective use of key information in patient care. It is true, however, that certified and trusted digital tools are needed [11]. Many digital FHH tools exist, however validation efforts are lacking to determine their utility in clinical care [17,18,19]. Because of health literacy capabilities in diverse and vulnerable health populations, some tools may not be preferred.

A virtual counselor with a speech interface may work better compared to a typical build your own pedigree in an FHH tool. A Spanish version of a virtual counselor with additional diseases is currently being developed [9]. Although context-dependent FHH tool applications will influence the choice of the users, a well-designed tool will have important implications for the quality of data collected and subsequent clinical utility.

### 1.4. Determining eHealth Quality

The International Organization for Standardization (ISO) is a worldwide federation of national standards bodies [20]. The ISO has set international requirements for health software related to product safety and lifecycle processes. The new ISO/TS 82304-2 technical specification on health and wellness app quality was developed by ISO Technical Committee (TC) 215, CEN TC 215, and IEC Subcommittee (SC) 62A under CEN lead.

The technical specification outlines ways for app developers and app assessment organizations to communicate the quality and reliability of health and wellness apps [21,22]. However, ISO/TS 82304-2 defines a set of 81 questions and supporting evidence usable to clarify the quality and reliability of a health app requiring broad expertise and technical knowledge. Not all these questions are relevant to primary care clinicians that work in FHH.

### 1.5. The FHH Quick App Review

The authors of this manuscript have extensive expertise in digital applications, validation requirements, and applying FHH in many clinical specialties, especially in primary care. We felt this was an opportunity to present new ideas and concepts from our *FHH Quick App Review* (defined in detail in the following sections) to the current FHH knowledge base. The purpose of this manuscript is to outline an abbreviated quality assessment process based on ISO/TS 82304-2, assisting primary care professionals in choosing quality FHH digital tools purposely for patient care, and specifically optimized for any clinical context encountered.

## 2. Materials and Methods

### 2.1. Health App Quality Scores

The new health app quality TS (ISO/TS 82304-2) proposes a set of 81 elemental product questions for app developers and manufacturers. The quality requirements and health app quality score calculation method have been developed using a Delphi consensus study [23]. Further input was gathered through surveys, interviews, and a review of existing standards and health app assessment frameworks. The health app quality label was tested with people with low health literacy.

Based on the answers and screenshots, a health app qualifies for a quality label if 4 minimum requirements are met: 5.2.2.1, 5.2.2.5, 5.2.4.5, and 5.4.1.1.4 with a score on four topics: healthy and safe, easy to use, secure data, and robust build. The results lead to an overall score on the label. The label allows users to compare apps and make an optimal choice. In the Netherlands, the National eHealth Living Lab (NeLL) had a role in the development of the ISO/TS 82304-2 and is currently working together with the Dutch healthcare field to implement it throughout the country. An image of the label can be found on the Netherlands standards website [24].

### 2.2. Application to Family Health History

While focusing on the subject of eHealth in Primary Care, we considered how a primary care physician with an interest in FHH could quickly review the quality and reliability of an FHH app, whether for clinical use, or use by their patients. Many of the ISO TS’s 81 questions are highly technical and/or unrelated to an instructive review by a physician analyzing specific FHH app capabilities.

Based on the many years of clinical and technical knowledge and experience of the authors, we recommend using our modified subset of 25 questions (known as the FHH Quick App Review) to get to the relevant knowledge of the functionality of an FHH app and its multifaceted benefits. Our ‘informative-only’ 25 questions do not produce a quality score but provide guidance to complete an initial but informed review of FHH apps. Another use of our questions is to filter out the questions and answers relevant to FHH in an already completed ISO/TS 82304-2 assessment.

The ISO standard’s scoring method highlights four categories. We list them here but include how they would apply to an FHH app.

Category 1: Healthy and safe (explains the medical value of using an FHH, limits health risks, makes clear the FHH app is not a diagnostic tool).

Category 2: Easy to use (clean graphical user interface for necessary functions e.g., create pedigree, enter health data for self and relatives, communicate with relatives and healthcare providers, share data, understand risk reports, etc).

Category 3: Secure data (follows general secure by design principles, security policy available to the user).

Category 4: Robust build (describes interoperability with other apps and systems, uses published standards for pedigree and family health history data, backs up data, can easily be maintained).

## 3. Results

Based on our review of the ISO/TS 82304-2 81 questions, the following list of 25 questions from the FHH Quick App Review in Table 1 is recommended to be more succinct as an initial review by primary care physicians who are specifically interested in FHH. Most of the questions are based directly on the ISO TS, some are modified or de novo. Where relevant, the original ISO question number is referenced to easily find the corresponding question, instruction, or answer in ISO/TS 82304-2, and ensure backward compatibility.

Even as the preceding questions are recommended, we want to stress that they should be viewed in the context of the purpose of the original ISO standard and its instructions. We encourage FHH app vendors to use the entire ISO quality label TS when possible, as the quality score it offers should be included in the final decision-making process.

To be complete, two other questions that some clinicians may find essential are: (1) are health professionals involved in the development of the FHH app? and (2) is appropriate peer-reviewed scientific literature used in the development of the app?

## 4. Discussion

Although we have presented general issues related to FHH adoption in this paper, it is more about determining FHH quality than about the implementation of eHealth in general, neither its potential barriers nor the adoption of eHealth apps or eHealth in general. We decided not to elaborate on this further in this particular manuscript as it could confuse, but the authors are indeed aware of the many opportunities available and the challenges to be overcome for the uptake of eHealth in primary care to ensure safe and high-quality development and implementation in primary care [25]. 

### 4.1. The FHH Quick App Review as a New Skill

A brief scan of the ISO quality label and score certainly offers an easier assessment of an FHH health app than taking the time to review the list of 25 questions in the FHH Quick App Review. The obvious limitation of using the Review is time spent evaluating each question with no quality score at the end. However, we believe considering each of the questions just once will provide the primary care professional with insightful knowledge and a deeper understanding of the benefits, risks, features, and limitations, of a quality FHH app, with a sharper eye toward key capabilities and desired usability. The clinician should then be proficient enough in subsequent FHH app reviews without going through all of the 25 questions again. 

### 4.2. Many Types of FHH Apps

Although this paper focuses on the evaluation of FHH apps by primary care physicians, it should be noted that in the realm of FHH data collection and use, there are many diverse users and stakeholders. FHH apps can be developed for several types and abilities of users, and for a wide range of use cases (see Figure 1). As this ecosystem emerges, interoperability issues will arise as to how we store and read from a single FHH record for each patient and family—no matter the app type or focus.

Key to this ecosystem are FHH apps linked to the patient’s medical record and genomic record (the latter repository containing whole genomic, large gene panel, pharmacogenomic, Single Nucleotide Polymorphism (SNP) microarray, genotyping, and sequencing data). This combination of data would advance the insights of clinical decision support systems. FHH apps and the algorithms used for risk assessment should follow clinical guidelines established by medical organizations. New guidelines are always under development, such as using polygenic risk scores in combination with FHH information. Guidelines on how to design clinical tools that act upon patient-reported data are still not fully developed.

When FHH is collected by primary care doctors, the focus is usually around 15 heritable diseases considered to be clinically actionable (Alzheimer’s disease, arthritis, asthma, breast and ovarian cancer, colon cancer, depression, diabetes, glaucoma, heart disease, high cholesterol, hypertension, kidney disease, osteoporosis, prostate cancer, and stroke). Of course, however, there will be other clinical uses and indications.

### 4.3. Use in Implementation

We believe the FHH Quick App Review is not only suitable for reviews but could also aid app development and implementation. A successful launch of any app by a healthcare provider requires an implementation plan. The value of the chosen app increases with data interoperability between apps, other data repositories, provider medical systems, and even genealogy systems.

This brings new opportunities but also some considerations. For example, we must be careful that we do not create more regulatory burdens for small and medium-sized clinics and health systems. The possibility of increased costs for quality labels in licenses and certifications can restrict the access to market for new developers, hampering innovation. The lack of new product development and competition reduces quality. Therefore, we used a subsection of an existing quality label in order to avoid the creation of an entirely new one.

Quality labels can also increase the price of certain apps as they require more rigorous quality controls. This can make them inaccessible to low-income areas. We feel the FHH Quick App Review prevents these problems by being open-source (no license cost) and understandable (no technical expertise required). It is also multi-use, as we envision these questions can also be used in FAQs and presentations. This might seem trivial, but in this emerging part of the medical space (genomics in primary care), having a tool that helps with the communication with physicians and that is dedicated to this specific topic can be invaluable for companies seeking to develop clinical-grade products.

In addition, it is important for the ecosystem that data sharing and consent systems for FHH apps do not exist. There are no shared FHH risk algorithm repositories either. Nevertheless, we should continue to ask for them. While the perfect FHH app may not exist today, we believe the knowledge to build a better, more capable FHH app is available. To do so, however, will require many stakeholders to come together. Our ‘informative-only’ 25 questions do not produce a quality score but do provide a guide to completing an initial review of FHH apps. Validation of the FHH App Quick App Review would imply using the Review in more specific contexts, such as country, medical system, financial organization, software/electronic health record, and competence-specific conditions for example. We preferred to present an FHH Quick App Review based on the new ISO/TS 82304-2 Technical Specification in a reductionist format in order to inform further research and potentially help in developing apps and clinical implementation.

### 4.4. Future Work

Besides offering the FHH Quick App Review as a solution to help determine FHH app quality, it should be noted that many other activities also focus on building better FHH apps. These groups are potential partners to test and validate the Review, making important contributions to a shared effort. Ongoing work could include the following:Promote ISO/TS 82304-2 beyond the EU. Many other countries still do not mention the quality label on their national websites.Industry strategies to reduce implementation gaps in family health history (see Table 1 in paper with the items listed under *Gap: effectiveness of digital tools and interoperability with and among EMRs:*) [26].The Global Alliance for Genomics and Health (GA4GH) is close to publishing its *Recommendation of a Common Data Set for FHH*, which includes a list of data elements that every FHH app should collect.The Global Genomic Medicine Consortium Family Health History Flagship Project piloting clinical FHH implementations around the world.Build consent systems and shared FHH risk algorithm repositories for FHH apps. Consent systems for FHH apps using locker-type systems as consent management. For example, the PGL project [27].Implement an FHH tool that is interoperable with other (EMR) systems. For example, through lockers and European Health Data Spaces [28].Case studies for finding out if and how physicians will use the FHH Quick App Review, which should be part of general eHealth education and integrated into vocational training and continuous professional development programs [29].With multiple stakeholders collecting and storing FHH data separately, the most efficient model would be a shared family-centric record, with access controlled by each family member to support their own needs [30].Include family medication history and pharmacogenomic results in the FHH pedigree to further optimize medication therapy [16].

## 5. Conclusions

From the perspective of the primary care physician, the world of eHealth is both medicinal and a scourge. Trying to determine the best FHH electronic tools for patient care and patient use can be time consuming. We offer the FHH Quick App Review as a possible solution. After going through the 25 review questions once for an FHH app, we believe it will result in a sharper eye for quality in other FHH apps and be a trusted guide for informed decisions. We also view the FHH Quick App Review as an achievable strategy to help improve app functionality and reduce implementation gaps. 

We look forward to the everyday use of the full ISO quality label TS for a comprehensive score that enables the comparison of multiple FHH apps, and a comparison of all other clinical and patient health apps. Not only will the application of these methods make the FHH ecosystem a reality, but also benefit the larger eHealth ecosystem for precision medicine and beyond.

## 6. Patents

No patents. FHH Quick App Review is based on ISO/TS 82304-2, which requires a license from the ISO.

## Figures and Tables

**Figure 1 genes-13-01407-f001:**
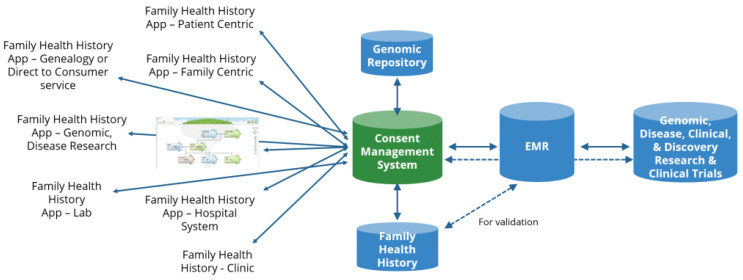
FHH app ecosystem. Image sourced from the Global Genomic Medicine Consortium. EMR: Electronic Medical Record.

**Table 1 genes-13-01407-t001:** FHH Quick App Review 25 questions.

Question Number	FHH Quick App Review Question	Derived from ISO/TS 82304-2 Question Number *
**1**	Who are the intended users of the FHH app? (Health professionals, patients, researchers, genetic/genomic/testing labs)	5.2.1.1
**2**	Are potential customers and users provided with adequate product information about the FHH app? (Full pedigree functionality or just a form, general or single disease focus, and use of personal data by the app, etc.)	5.3.2.5
**3**	What is the intended use or purpose of the FHH app? a./Inform–learning the value of FHH in genetic-based care. b./Simple monitoring–personal health tracking that may benefit both the patient and their family members. c./Communicate–between family members, between patient and doctor or genetic counselor, and other health services. d./Research–consenting and sharing FHH data and information with one or many research organizations. e./Calculate–FHH app generates a heritable risk assessment, a percentage risk of a genetic mutation for a relative yet to be sequenced, and relatives indicated for cascade testing. f./Diagnose–determine heritable risk, provide validated information and clinical interpretation on genetic variants, recommend appropriate genetic/genomic testing, all under physician and genetic counselor guidance (see/Communicate). g./Treat–order, schedule, and track testing, clinical use of genomics and pharmacogenomics, initiating indicated clinical surveillance, and genetic counseling. h./Preventative behavior change–makes recommendations and tracks lifestyle changes. i./Self-manage–provides tools for patients who collect and examine their medical, ‘Omic, and family health history data, along with patient-generated data (wearables, other devices).	5.2.1.4
**4**	For which health issue(s) and/or health need(s) is the FHH app intended? (a specific gene, disease?)	5.2.1.3
**5**	In which languages is the FHH app available?	5.1.1.4
**6**	Are the health risks of the health app analyzed? (Health risks may include over-reliance on the app, misinterpretation of information, medical judgments made without input from a medical professional, etc.)	5.2.2.1
**7**	Are measures in place to control the health risks of the FHH app? (Is there information covering user safety, including warnings and limitations of use? Is user training offered? Is there documented verification and validation of risk reports and algorithms used by the app, etc.)	5.2.2.2
**8**	Are the residual risks of using the health app found to be acceptable? Is a consent system in place for patient privacy and security (data sharing options explained, data hiding or limiting options available with family members, healthcare providers, medical systems, protecting paternity confidences, how is de-identified data created, etc.)?	5.2.2.3
**9**	Is a process in place to collect and review safety concerns and incidents for the FHH app? (how can users report incidents or issues)?	5.2.2.6
**10**	Are ethical challenges of the health app assessed with both health professionals and intended users in mind? (information not used for health insurance qualification, being sensitive to cultural issues, how to report medically actionable health risks without causing undue concern, understanding the role of biological relationships vs. non-biological and chromosomal sex for risk assessment purposes, respecting the privacy of the user’s family members, etc.)?	5.2.3.1
**11**	Describe the health benefits of using the FHH app. Are potential users made aware of the health interventions applied to achieve the health benefits? Is there a review of possible interventions that could be ordered or performed by the Primary Care physician?	5.2.4.1, 5.2.4.2
**12**	Are potential users made aware of the need for support of a health professional to achieve the health benefits?	5.2.4.2
**13**	Are potential users made aware of the financial costs to achieve the health benefits? (out-of-pocket costs for laboratory testing and other interventions)	5.2.4.3
**14**	Are there maintenance processes for the health information in the FHH app by the app developer? Are all sources for the health information disclosed to users?	5.2.4.6
**15**	Does the FHH app encourage and track the user by keeping the pedigree and FHH information up to date?	Not derived from ISO
**16**	Is the design of the FHH app driven and refined by user-centered evaluation? Is the app design based on an explicit understanding of users, tasks, and environment?	5.3.2.3
**17**	Are measures in place to avoid user error and reasonably foreseeable misuse of the health app? (Input error detection, context-sensitive help, etc.)	5.3.2.4
**18**	Are instructions for use readily available for users? Are appropriate resources available to adequately help users who experience problems with the app?	5.3.2.6, 5.3.2.7
**19**	Is a privacy statement readily available to potential users of the FHH app? Is an appropriate retention policy established to erase or review the data stored?	5.4.1.1.3, 5.4.1.1.4
**20**	Is a secure by design process followed? (Security by design ensures that information security is designed and implemented within the development lifecycle of the app.)	5.4.2.3
**21**	Is user authentication, authorization, and session management implemented to secure access to the health app? (The identity of an app user is authenticated prior to access of any personal identifying information.)	5.4.2.7
**22**	Are security vulnerabilities reported, identified, assessed, logged, responded to, disclosed, and quickly and effectively resolved?	5.4.2.9
**23**	Is the information security policy readily available to potential users?	5.4.2.11
**24**	Is data interoperability achieved via published FHH data standards like HL7 FHIR (Fast Healthcare Interoperability Resources) and GA4GH (Global Alliance for Genomics and Health)? Are certified interoperable systems listed (what other apps, data repositories, and major medical systems does the app communicate with)?	Not derived from ISO
**25**	Can users obtain or share their health-related PII by a data import/export from/to another FHH app or platform? Can the app import pedigree data from other genealogical services?	5.5.2.4

* Access to the derived question number from ISO/TS 82304-2 is only available after the purchase of a license. See https://www.iso.org/standard/78182.html (accessed 1 June 2022).

## Data Availability

Not applicable.

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
