# Peer review of "FHH Quick App Review: How Can a Quality Review Process Assist Primary Care Providers in Choosing a Family Health History App for Patient Care?"

_genes, 2022, doi:10.3390/genes13081407_

Round 1
Reviewer 1 Report
Your manuscript is well -structured and the arguments follow in a logical order.
Having said that, the following aspects could be further improved:
- Has such a 'reductionist' approach been implemented before on eHealth apps (beyond FHH)? It would be good to know if successful examples exist elsewhere.
- The selection of the 25 questions is based on personal expertise, have they been validated?
- Is there a case study or model application of the recommended questionnaire available? i.e. has it been tested at all?
- While you mention potential barriers for the adoption of eHealth apps into routine practice in primary care, your discussion does not directly address such identified aspects from previous studies.
- Please provide 1-2 sentences of potential limitations of this recommended approach.
Author Response
Reviewer #1
Reviewer 2 Report
Dear authors,
thank you for the the interesting article. I can only comment from my GP perspective.I am not very familiar with process evaluations.
Abstract
There is too much space between line 12 and line 13
Methods
Could you drescribe more in detail how you manage to reduce the 81 to 25 questions? How many GPs where involved in the selection-process? How was it done? Could you add another figure or flowchart to make it clearer?
Why did you not involved patients in this process? Please adress this in the discussion.
Results
P. 7, question 24: What means „HL7 FHIR and GA4GH?“
P 8 line 197: what means „SNP“?
P 8 line 203: the information about the diseases are mentioned too late, please put them into the introduction.
Discussion
It remains a bit unclear, if patients or GPs should be the future users. How should the app be designed, that disadvantaged groups can also use them? What about patients who are not so tech-savvy (geriatrics, dementia, migrants...)?
Figure 1: What means EMR?
Author Response
Reviewer #2

Round 2
Reviewer 1 Report
Thank you for addressing the reviewer's concerns.